# Precise Calibration of the Continuous Surface Cap Model for Concrete Simulation

**Yury Vladislavovich Novozhilov** [1,*] **, Andrey Nikolaevich Dmitriev** [1] **and Dmitry Sergeevich Mikhaluk** [2]

1    Peter the Great St. Petersburg Polytechnic University, 195220 Saint-Petersburg, Russia;
     dmitriefan@outlook.com
2    Center of Engineering Physics Simulation and Analysis (CEPSA JSC), 191197 Saint-Petersburg, Russia;
     dmitry@multiphysics.ru
*    Correspondence: yury.novozhilov@outlook.com; Tel.: +7-(921)-309-30-51

**Abstract:** The Continuous Surface Cap Model (CSCM) is one of the most widely used concrete models in LS-DYNA. The model is capable of capturing many important nonlinear mechanical behaviors of concrete well. The model has a built-in auto calibration procedure based on CEB-FIP code data. However, the built-in calibration procedure estimates material properties with significant errors, especially for tensile strength. Our study highlights the imperfection of the built-in automated material calibration procedure by the example of one-element uniaxial tension and compression tests. A calibration procedure is proposed, which significantly improves the accuracy of the material properties calculation: tensile and compressive strength and fracture energy. It is shown that the model with the proposed calibration procedure can describe the structure defamations and the fracture zone patterns more accurately.

**Keywords:** concrete; constitutive model; numerical simulation; LS-DYNA; CSCM; impact; progressive collapse; fracture; damage

## 1. Introduction

Ansys LS-DYNA finite element (FE) code provides more than ten internal constitutive models specially developed for concrete material simulation. Among them you can find such models of concrete as Holmquist–Johnson–Cook (HJC), Riedel–Hiermaier–Thoma (RHT), Karagozian&Case concrete (KCC) and Continuous Surface Cap Model (CSCM). The descriptions of the theory behind these models could be found in software user manuals [1] and publications of many applied researchers [2–5].

The CSCM (Continuous Surface Cap Model) [6,7] model, considered in this paper, is based on Frank L. DiMaggio's work [8], published in 1971. The current CSCM model implementation in LS-DYNA is calibrated according to CEB-FIP 1990 Model Code [9]. The material model is developed at the request of the Federal Highway Administration of the U.S. Department of Transportation.

The CSCM model implementation in LS-DYNA has numerous essential features that simulate concrete material mechanics with a high level of accuracy. The model uses isotropic constitutive equations and three stress-invariant strength surfaces with translation for pre-peak hardening, and a hardening cap that expands and contracts. Independent tensile (brittle) and compressive (ductile) damage-based softening tracking allows simulating virtual crack closings in compressive stress or strain states. The rate effects for high strain rate applications influence material strength and fracture energy release estimation. The model has a built-in energy regularization mechanism that reduces mesh sensitivity. Material erosion is supported for FE simulation of perforation and scabbing; the model also supports meshless particle discretization methods, such as Smoothed Patrice Hydrodynamics (SPH) and Smoothed Patrice Galerkin (SPG).

Moreover, the model has an easy input regime that activates an internal auto model calibration procedure. The model could automatically generate all material input paraments based on unconfined compressive strength and average aggregate size due to easy input regime. This capability is essential since it can simulate concrete objects on the design stage when no experimental data on material properties are available. This internal calibration works for concrete with unconfined strength is 20–58 MPa, but the best accuracy is achieved for the range of 28–48 MPa [6]. The fracture energy calculation works correctly for a characteristic aggregate size range of 8–32 mm.

Due to all advantages mentioned above, the CSCM model has been widely used in simulations of concrete structures subjected to drop-weight impact [10–12], projectile penetration [13–15], progressive collapse [16–20], vehicle collision [21–23] and explosion [24–27].

Numerous studies have shown a good agreement between numerical results and experimental data. Bermejo et al. [17] conducted an accurate scale test of a two-floor structure that loses a penultimate bearing column; a related numerical validation with the CSCM concrete model showed actual displacements and construction failure modes. Qian et al. [28] found a good match of displacements, crack patterns, and a small mesh size dependency in the simulation of RC slab under a two-column loss scenario. Zhang et al. [25] simulated the simply-supported reinforced concrete (RC) beams subjected to the combination of impact and blast loads and obtained correct vertical displacements, reaction forces and crack distribution. Yu et al. [19] quasi-statically investigated the effect of masonry infill walls on the progressive collapse resistance of RC frames. Results obtained from numerical simulation and experiments agree in non-ultimate loading levels. Grunwald et al. [29] simulated column loss for a two-dimensional frame structure under blast load and confirmed that the vertical displacement from the numerical model is in good agreement with test data, but crack patterns are not described correctly.

At the same time, many authors have noted an imperfect correlation with experiments when using default parameters of CSCM. Kim S.B. et al. [10] used the auto-generated CSCM parameters to simulate the RC beams under drop weight loading conditions. They concluded that the strength is overvalued, and the vertical displacement is underestimated. Numerical models of missile impacts on RC plates developed by Chung et al. [15] overestimated the residual displacements and rebound in bending impact tests and showed a more significant maximum displacement in a punching impact test.

Many authors have attempted to calibrate the model manually for a more accurate description of concrete structures subjected to dynamic actions. Levi-Hevroni et al. [30] based experiments on the tension split Hopkinson bar, and suggested an increase in fracture energy and parameters governing the strain rate effects, since the numerical results did not correspond well with the test data. Yu et al. [20] found that default model parameters caused stiffer and more significant resistance of RC beam-slab substructures under perimeter column loss. The reduction of elastic modulus and fracture energy allowed them to get a good match in cracking and displacements.

Thus, the CSCM model with automatically generated parameters can lead to incorrect simulation results for RC structures. As mentioned above, many researchers have attempted to calibrate the material model parameters more accurately. However, these studies are fragmentary, and to date there is no unified methodology for calibrating the CSCM model.

The object of the study in this paper is the concrete material model CSCM implemented in LS-DYNA as *MAT_CSCM(_CONCRETE)/*MAT_159 card. The goal of this research is to create a model calibration methodology and to validate the developed methodology on two problems of dynamic deformation of reinforced concrete with known experimental data: under low-velocity impact [23] and under progressive collapse [31].

The proposed procedure for model parameter identification and calibration is assumed that all input parameters will be identified on the material density, cylindrical strength, and fracture energy/characteristic aggregate size. Other input parameters are calculated based on a combination of the relations presented in [6,9,32,33]. A material model calibrated in

this way should show more accurate compliance with strength standards [32,33] and work for a broader range of concrete strength classes.

## 2. Materials and Methods

### 2.1. A Mechanics Problem

Concrete is a strongly heterogeneous material that consists primarily of aggregate and mortar and exhibits a complex nonlinear mechanical behavior. Although it has cracks and discontinuities on a microscopic scale, the stress-strain relationships of concrete in the CSCM are described with classical equations of continuum mechanics. The following subsections will show a summary of these classical relations, such as equation of motion and conservation laws.

#### 2.1.1. Equation of Motion

Consider the deformation in time of an arbitrary fixed volume $\Omega_0$ bounded by a smooth, closed surface $\Gamma = \Gamma_1 \cup \Gamma_2 \cup \Gamma_3$. The equation of motion can be written as:

$$\rho \ddot{x}_i = \sigma_{ij,j} + \rho f_i \tag{1}$$

Three types of boundary conditions can be imposed on Equation (1) in the general case:

- At the boundary $\Gamma_1$, essential boundary conditions: $\sigma_{ij} n_j \big|_{\Gamma_1} = \tau_i(t)$;
- At the boundary $\Gamma_2$, natural boundary conditions: $u_i(t) \big|_{\Gamma_2} = U_i(t)$;
- At the boundary $\Gamma_3$, contact boundary conditions: $\left( \sigma_{ij}^+ - \sigma_{ij}^- \right) n_j \big|_{\Gamma_3} = 0$.

where $\sigma_{ij}$ is stress tensor, $f_i$ is volume force, $\ddot{x}_i$ is acceleration, $n_j$ is external normal for boundary, $\sigma_{ij}^+$, $\sigma_{ij}^-$ is stress tensors for bodies in contact and $u_i$ is displacement vector.

#### 2.1.2. Conservation Laws

Mass conservation law can be written in the form:

$$\rho V = \rho_0 V_0, \tag{2}$$

where $\rho_0$, $\rho$ is initial and current density and $V$, $V_0$ are initial and current volume, respectively.

The energy conservation law has the form:

$$\dot{e} = \Theta s_{ij} \dot{\varepsilon}_{ij} - p \dot{\Theta}, \tag{3}$$

where $s_{ij}$ is stress tensor deviator, $\dot{\varepsilon}_{ij}$ is strain rate tensor, $p$ is pressure, $\Theta$ is specific volume and $e$ is internal energy per unit volume.

The deformations and displacements are related through geometric relations:

$$\varepsilon_{ij} = \frac{1}{2} \left( u_{i,j} + u_{j,i} \right). \tag{4}$$

To close the written equations, it is necessary to set physical relations linking stresses, strains, strain rates, temperature, etc. In this paper, the CSCM material model is used to describe the deformation of concrete material, the physical relations of which are described below.

### 2.2. Concrete Material Model

The subsection provides information about the CSCM mathematical model features. The section content is a synthesis of the materials presented in [6,7,34].

#### 2.2.1. Elastic Behavior

In the elastic region, concrete is considered isotropic, and Hooke's Law is used for the elastic stress–strain relationship.

### 2.2.2. Strength Surface

Within this model, the strength surface has the form:

$$f(I_1, J_2, J_3, \chi) = J_2 - \omega^2 F_f^2 F_c, \tag{5}$$

where $I_1 = \sigma_{ii}$ is the first invariant of the stress tensor, $J_2 = \frac{1}{2}s_{ij}s_{ij}$ is the second invariant of the stress tensor deviator, $J_3 = \frac{1}{3}s_{ij}s_{jk}s_{ki}$ is the third invariant of the stress tensor deviator, $F_f$ is meridional surface, $F_c$ is elliptical cap surface, $\omega$ is Rubin's scaling function and $\chi$ is cap surface hardening parameter. A view of the strength surface is shown in Figure 1d.

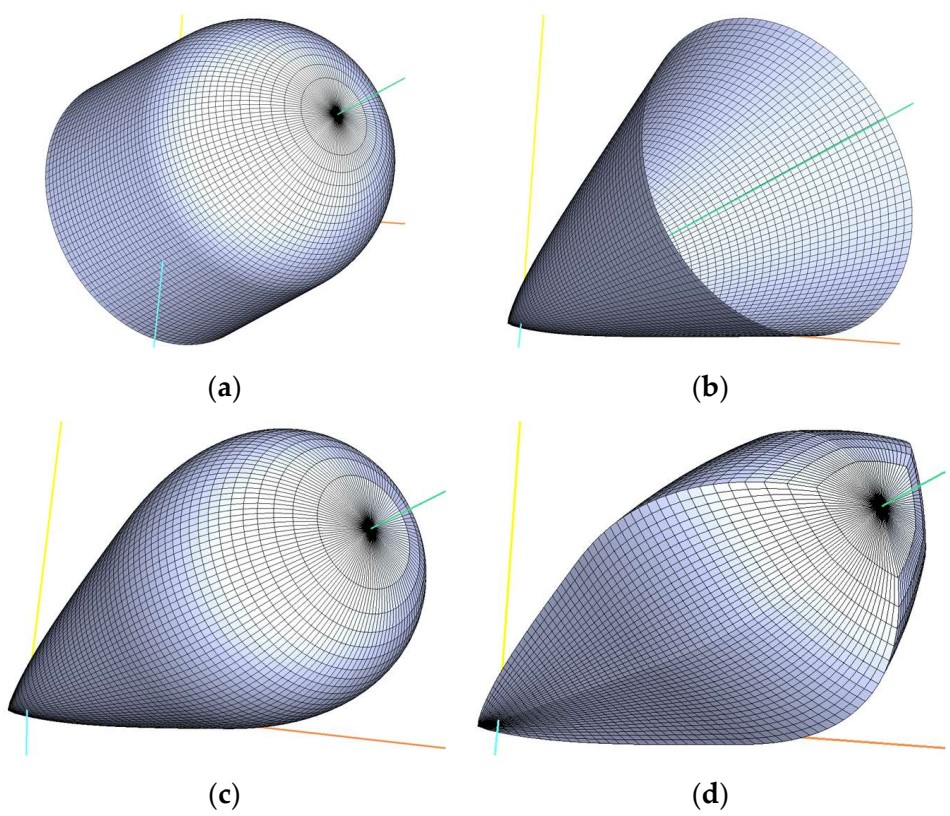

**(a)**                                                          **(b)**

**(c)**                                                          **(d)**

**Figure 1.** The general shape of concrete model yield surface in the Principal Stress Space (PSS): (**a**) the triaxial compression, $J_2 - F_c = 0$; (**b**) the shear surface, $J_2 - F_f^2 = 0$; (**c**) combined surface, $J_2 - F_f^2 F_c = 0$; (**d**) complete strength surface, $J_2 - \omega^2 F_f^2 F_c = 0$.

### 2.2.3. Triaxial Compression

The stresses calculated from the elastic relations are hereinafter referred to as the trial elastic stresses where $\sigma_{ij}^T$, denotes their corresponding invariants $I_1^T, J_2^T, J_3^T$. The material behaves elastically when $f(I_1^T, J_2^T, J_3^T, \chi_0) \leq 0$. When $f(I_1^T, J_2^T, J_3^T, \chi_0) > 0$, the material begins to behave elastically–plastically, in which case the algorithm returns the stresses to the strength surface, so that $f(I_1^P, J_2^P, J_3^P, \chi) = 0$, wherein this case the associated flow law applies. Meridional surface $F_f$ is described by the following equation:

$$F_f(I_1) = \alpha + \theta I_1 - \lambda e^{-\beta I_1}, \tag{6}$$

where $\alpha$, $\beta$, $\theta$, $\lambda$ are material parameters obtained from triaxial compression tests (TXC) on concrete cylinders. The general shape of concrete model yield surface in the Principal Stress Space (PSS) is shown in Figure 1a.

### 2.2.4. Cap Surface

The elliptical surface $F_c$ describes the change in volume due to collapsing pores in concrete. This surface has the form:

$$F_c(I_1, \chi) = 1 - \frac{(I_1 - L(\chi))(|I_1 - L(\chi)| + I_1 - L(\chi))}{2(X(\chi) - L(\chi))^2}, \tag{7}$$

where $L(\chi)$ is defined as:

$$L(\chi) = \begin{cases} \chi & \text{for } \chi > \chi_0 \\ \chi_0 & \text{for } \chi \leq \chi_0 \end{cases}. \tag{8}$$

The equation for $F_c$ at $I_1 \leq L(\chi)$ equates to one, at $I_1 > L(\chi)$ it describes an ellipse. The cap surface and the lateral surface intersect at $I_1 = \chi$. $\chi_0$ is the value of $I_1$ at the intersection of the cap and the lateral surface before the beginning of hardening (before the cap moves).

Intersection of the cap with the PSS hydrostatic axis occurs at the point $I_1 = X(\chi)$. This point depends on the ellipticity parameter $R$:

$$X(\chi) = (\chi) + RF_f(L(\chi)). \tag{9}$$

The cap movement simulates a plastic change in volume. When the cap expands ($X(\chi)$ and $\chi$ increase), it simulates volume contraction; when it shrinks, it simulates volume expansion, i.e., dilatation. The cap movement is subject to the following law of hardening:

$$\varepsilon_V^P = W\left(1 - e^{-D_1(X-X_0) - D_2(X-X_0)^2}\right), \tag{10}$$

where $\varepsilon_V^P$ is the plastic volume strain, $W$ is the maximum plastic volume strain, $D_1$ and $D_2$ are material parameters and $X_0$ and $X$ are the initial and current points of hydrostatic axis with the cap intersection, respectively.

The shear surface in the PSS is shown in Figure 1b. The combined surface $J_2 - F_f^2 F_c = 0$ is shown in Figure 1c.

### 2.2.5. Triaxial Extension and Torsional Conditions

It has been experimentally established that concrete begins to fail in triaxial tensile and torsional conditions at lower values of $J_2$ than in three-axis compression. This fact suggests that the strength surface depends on the third invariant of the stress tensor deviator $J_3$. The three-invariant strength surface takes the form of a triangle or hexagon in the deviator plane.

### 2.2.6. Scaling Function

The scaling function $\omega$ introduces the dependence of any stress state at the strength surface on the triaxial compression stress state as $\omega F_f$. $\omega$ depends on the angle $\hat{\beta}$, which varies within $-\frac{\pi}{6} < \hat{\beta} < \frac{\pi}{6}$ and is expressed through $J_2$ and $J_3$ as:

$$\sin 3\hat{\beta} = \hat{J}_3 = \frac{3\sqrt{3}J_3}{2J_2^{3/2}}, \tag{11}$$

where $\hat{J}_3$ is a normalized invariant, whose values lie within $-1 < \hat{J}_3 \leq 1$. In this case, for three-axis compression $\hat{J}_3 = 1$, for torsion $\hat{J}_3 = 0$ and for triaxial tension $\hat{J}_3 = -1$.

The equations for determining the scaling function $\omega$ are as follows:

$$\begin{cases} \omega = \frac{-b_1 + \sqrt{b_1^2 - 4b_2 b_0}}{2b_2}; \\ b_2 = (\cos\hat{\beta} - a\sin\hat{\beta})^2 + b\sin^2\hat{\beta}; \ b_1 = a(\cos\hat{\beta} - a\sin\hat{\beta}); b_0 = -\frac{(3 + b - a^2)}{4}; \ b = (2Q_1 + a)^2 - 3; \\ a = \frac{-a_1 + \sqrt{a_1^2 - 4a_2 a_0}}{2a_2}; \ a_2 = Q_2; \ a_1 = \sqrt{3}Q_2 + 2Q_1(Q_2 - 1); \ a_0 = 2Q_2^2(Q_2 - 1). \end{cases} \tag{12}$$

The functions $Q_1$ and $Q_2$ are functions of $I_1$, which allows the section of the strength surface of the deviator plane to change from a triangle to an irregular hexagon and a circle as the pressure increases. The dependence of $Q_1$ and $Q_2$ on $I_1$ is as follows:

$$Q_1 = \alpha_1 - \lambda_1 e^{-\beta_1 I_1} + \theta_1 I_1, \tag{13}$$

$$Q_2 = \alpha_2 - \lambda_2 e^{-\beta_2 I_1} + \theta_2 I_1, \tag{14}$$

where $\alpha_1$, $\lambda_1$, $\beta_1$, $\theta_1$ и $\alpha_2$, $\lambda_2$, $\beta_2$, $\theta_2$ are material parameters. $\omega = Q_1 F_f$ in triaxial tension and $\omega = Q_2 F_f$ in torsion.

The functions $Q_1$ and $Q_2$ control the shape of the surface only at compressive loads. At tensile loads these functions take the values $Q_1 = \sqrt{3}/3$ and $Q_2 = 0.5$, which leads to a triangular section of the limiting surface by a deviator plane.

### 2.2.7. Damage

Concrete exhibits softening (strength reduction) in the tensile and low to moderate compressive regimes, which is simulated by introducing damage as:

$$\sigma_{ij}^d = (1 - d)\sigma_{ij}^{vp} \tag{15}$$

The damage formation is applied to the stresses after they are updated by the viscoplasticity algorithm. Here, $d$ is a scalar damage parameter that transforms the stress tensor without damage $\sigma_{ij}^{vp}$ into the stress tensor with damage $\sigma_{ij}^d$. Damage $d$ can increase from 0 to 1 and is subdivided into brittle and ductile damage. The damage starts to accumulate when the strength surface reaches – damage accumulation for $f_c = 50$ MPa concrete is shown in Figure 2.

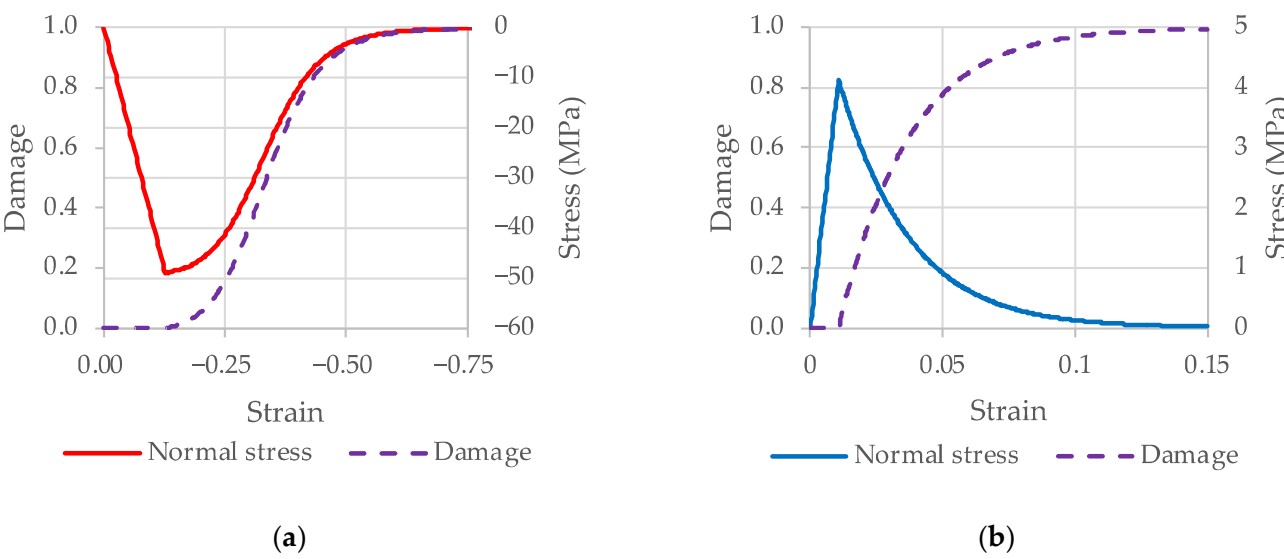

(**a**)                                                        (**b**)

**Figure 2.** Damage accumulation for $f_c = 50$ MPa: (**a**) Compression, (**b**) Tension.

Ductile damage begins to accumulate when the pressure $P$ is compressive (positive) and the value of term $\tau_c = \sqrt{\frac{1}{2}\sigma_{ij}\varepsilon_{ij}}$ exceeds the threshold value $\tau_{0c}$. The ductile damage $d_c$ itself is determined by the equation:

$$d_c = \frac{d_{\max}}{B}\left(\frac{1+B}{1+Be^{-A(\tau_c-\tau_{0c})}} - 1\right), \tag{16}$$

Brittle damage begins to accumulate when the pressure $P$ is tensile (negative) and the value of the term $\tau_t = \sqrt{E\varepsilon_{max}^2}$ exceeds the threshold value $\tau_{0t}$. Here, $\varepsilon_{max}$ is the maximum

principal deformation, where $E$ is the Young's modulus of undamaged concrete. The damage is determined by the equation:

$$d_t = \frac{0.999}{D} \left( \frac{1+D}{1+De^{-C(\tau_t - \tau_{0t})}} - 1 \right),$$ (17)

where values $A$, $B$, $C$, $D$, $d_{\max}$ are material parameters.

The maximum current ductile or brittle damage, $d = \max(d_t, d_c)$, is chosen to substitute in Equation (3).

### 2.2.8. Strain Rate

The strain rate has a significant influence on the strength properties of concrete. As the strain rate increases, the material hardens. The effects are considered in viscoplastic form in the CSCM model [12].

The viscoplastic algorithm interpolates between elastic trial stresses $\sigma_{ij}^T$ and unviscous plastic stresses $\sigma_{ij}^P$ (without regard to rate hardening) at each time step to obtain viscoplastic stresses (including rate hardening):

$$\sigma_{ij}^{vp} = (1-\gamma)\sigma_{ij}^T + \gamma\sigma_{ij}^P, \ \gamma = \frac{\Delta t/\eta}{1+\Delta t/\eta}.$$ (18)

This interpolation depends on the effective yield factor $\eta$ and the time step $\Delta t$. The coefficient $\eta$ is calculated through the given material parameters using the equation for tension loads:

$$\eta = \eta_s + \left( \frac{-I_1}{\sqrt{3J_2}} \right)^{PWRT} (\eta_t - \eta_s),$$ (19)

and for compression loads:

$$\eta = \eta_s + \left( \frac{I_1}{\sqrt{3J_2}} \right)^{PWRC} (\eta_c - \eta_s),$$ (20)

where $\eta_s = SRATE \cdot \eta_t$, $\eta_t = \frac{\eta_{0t}}{\dot{\varepsilon}^{N_t}}$, $\eta_c = \frac{\eta_{0c}}{\dot{\varepsilon}^{N_c}}$, $\dot{\varepsilon}$ is strain rate; $SRATE$, $\eta_{0t}$, $\eta_{0c}$, $N_t$, $N_c$ and $PWRT$, $PWRC$ are the material parameters.

### 2.2.9. Material Model Calibration

The *MAT_CSCM_CONCRETE model could be initialized with only three mechanical parameters direct input: material density $\rho$, compressive cylindrical strength $f_c$ and average aggregate size $d_{\max}$. Thus, all other model parameters will be calculated with the so-called easy input initialization procedure.

Paper [35] proposed a calibration procedure for the *MAT_SCHWER_MURRAY_CAP_MODEL model, similar to the CSCM strength surface and cap surface. But the procedure could not be directly translated to the CSCM model, since *MAT_SCHWER_MURRAY_CAP_MODEL has different relations for calculating the fracture energy parameters and the material rate-dependent parameters. *MAT_SCHWER_MURRAY_CAP_MODEL cannot regularize the fracture energy to also reduce the influence of mesh effects on the damage accumulation.

It was thus decided to replace the original equations for strength surface with the equation proposed in [35]. All other data is generated based on revised actual equations [6,9,32,33]. The new proposed set of equations allows calculating *MAT_CSCM input parameters based on the same scope of input parameters: $\rho$, $f_c$ and $d_{\max}$. These changes are aimed at improving the accuracy of the strength properties of the material and a better description of its failure process.

The proposed set of equations is presented in the Appendix A of the paper.

### 2.3. Original Verification

The results of the CSCM model with the proposed external calibration must be verified with a full-scale simulation of concrete structures. Several studies on concrete and reinforced concrete (RC) structures under extreme loads [36] simulation with proposed CSCM calibration have been published already. Thus, simulation of low-speed impact on reinforced concrete beams [37], RC slab blast loading [38] and pane concrete slab perforation by a rigid missile with adaptive solid-to-SPH switching [13] was successfully done with the proposed CSCM calibration. All this research shows a good correlation between numerical results and experimental data. Despite a wide range of verification cases, there is no direct simulation results comparison between built-in auto and proposed external calibration for CSCM. This comparison is done in the current paper.

Two verification cases are selected for this purpose. The first one is the original verification from the Evaluation of LS-DYNA Concrete Material Model 159 manual [23]. This experiment is used for CSCM auto calibration fitting. It is expected that the new proposed calibration should not deteriorate the predictability of the model result.

The second case is related to a two-story frame progressive collapse [31]. This case is selected as an example worth lading scenario for an auto procedure. The frame concrete strength is on the border of the accuracy application range for built-in auto CSCM calibration. The collapse of the structure occurs mainly under tensile loads.

Both verification case simulations are performed with the pure finite element method (FEM) in the Lagrange equation. Since element erosion can introduce an additional unphysical fitting parameter into the calculation [39], the feature is always turned off. The concrete material softening could be simulated due to correct damage mechanism work only.

#### 2.3.1. Single Element Strength Estimation

Quasistatic single FE unconfined compression and tension tests are performed at the first stage of the model response studies. The boundary conditions for the model are shown in Figure 3, where $u(t)$, or the function of the constant rate of compression or tension of the model. This test should show the correctness of material peak strength estimation.

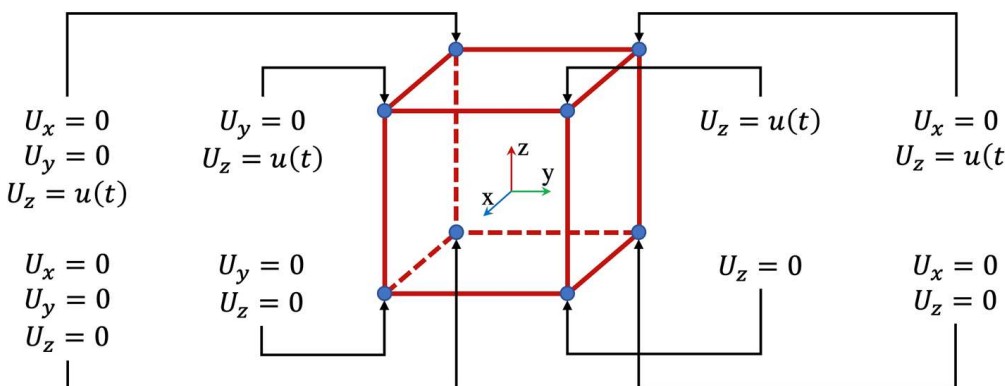

**Figure 3.** Single FE unconfined compression and tension tests illustration.

The peak compressive strength that *MAT_CSCM_CONCRETE model estimates is compared with the compressive cylindrical strength $f_c$, used for model initialization. The peak tensile strength $f_t$ could be estimated as shown in Equation (21) [9,32,33].

$$f_t = \begin{cases} 0.3(f_c)^{\frac{2}{3}}; \ f_c \leq 50 \, \text{MPa} \\ 2.12 \ln(1 + 0.1(f_c + \Delta f)); f_c > 50 \, \text{MPa} \end{cases} ; \tag{21}$$

where $\Delta f = 8$ MPa. The values $f_c$ and $f_t$ are compared with numerical results from tests with unconfined elements.

### 2.3.2. Single Element Fracture Energy Estimation

Concrete fracture energy $G_F$ on par with strength is one of the most critical material properties determining the material's post-peak behavior. The model should be able to show correct fracture energy response that agrees with the input data.

Default tensile fracture energy could be estimated as:

$$G_F = G_{F0}\left(\frac{f_{cm}}{f_{cm0}}\right)^{0.7}, \tag{22}$$

where $G_{F0}$ are the base values of fracture energy, as found in Table 1.

**Table 1.** Base values of fracture energy $G_{F0}$.

| $d_{\max}$ (mm) | $G_{F0}$ (N/mm) |
| --- | --- |
| 8 | 0.025 |
| 16 | 0.030 |
| 32 | 0.038 |

Base values of fracture energy from Table 1 could be interpolated with (4).

$$G_{F0} = 0.021 + 5.357 \cdot 10^{-4} d_{\max} \tag{23}$$

The fracture energy calculations are made for the single finite element tension. The average aggregate size $d_{\max}$ is set to 16 mm. $G_F$ is calculated as an area under the descending branch of the stress-crack opening curve and compared to CEBFIP 1990 [9].

### 2.3.3. Impact on the RC Beam

The original verification case set «4» with impact on over-reinforced concrete beams from the Evaluation of LS-DYNA Concrete Material Model 159 manual [23] is selected. The LS-DYNA FE models from set «4» are published freely by the solver developers [40]. This case contains a good set of experimental data and could be simulated without material erosion. The FE model sensitivity studies boundary conditions, mesh size and other solve process settings carried out in the original manual [23]. Thus, we can use the original developers' FE models from set «4» as is and focus only on the material model calibration influence on the results.

The case set contains three test cases with a deflection history measurement. The beams are impacted by two steel 32 mm diameter cylinders and supported on each end by 32 mm diameter cylinders. Basic model dimensions are presented in Figure 4.

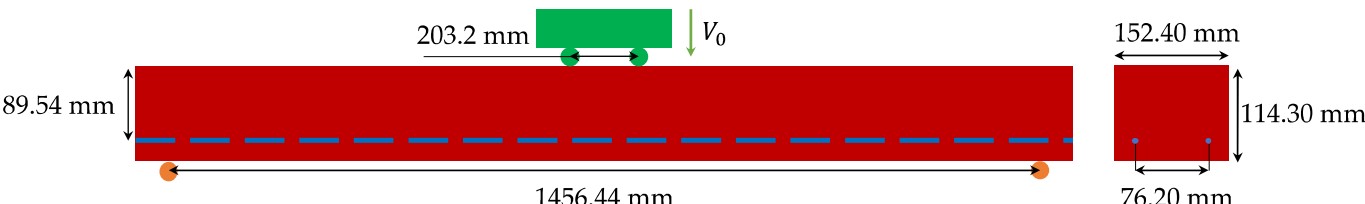

**Figure 4.** Basic model dimensions.

The nominal compressive concrete strength is 28 MPa and a 10 mm aggregate is assumed [23]. The stress–displacement behavior of the rebar was also measured. The initial yield strength of the steel reinforcement was 457 MPa, with an ultimate stress of about 614 MPa.

The concrete is modeled with 8.5 mm hex elements—the mesh size introduced in original verification models [40]. The reinforcement is modeled with beam elements with common nodes with the concrete hex elements. The Cowper–Simonds strain rate model

with parameters $C = 40 \, s^{-1}$, $p = 5$ is used for steel rebar viscous effect captured during dynamic loads [41].

The impactors and supporting cylinders are modeled with hex elements and assumed rigid. The model used in the simulation can be seen in Figure 5.

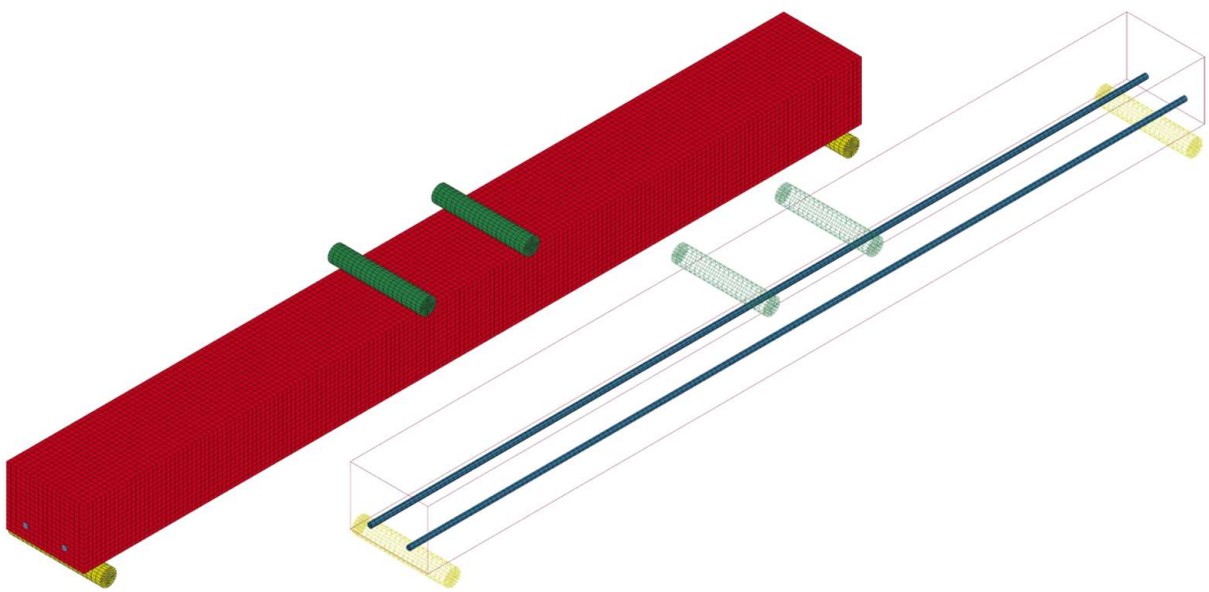

**Figure 5.** RC beam FE model overview.

Concrete interacts with rigid cylinders through segment-based symmetric contact with constant friction assumed as 0.45. Belytschko–Bindeman assumed strain co-rotational stiffness hourglass control is used with hourglass suppression coefficient 0.03.

Three test cases with different drop weight and impact speed are considered (Table 2). Drop weight changes modeled but rigid cylinder density changes.

**Table 2.** Load cases.

| Case Name | Drop Weight (kg) | Impact Velocity (m/s) |
| :---: | :---: | :---: |
| B | 31.75 | 7.3 |
| C | 47.86 | 6.0 |
| D | 63.93 | 5.2 |

2.3.4. Two-Story Frame Progressive Collapse

The reference experiment on progressive collapse is presented by removing a corner column of an RC two-story frame with loads, geometry and mechanical properties reflecting design conditions [31,42]. The bays above the removal column were loaded with concrete blocks imitating dead and live loads and the weight of outside walls. The cylinder compressive strength of concrete is about 30 MPa, and the yield strength for the whole reinforcement was 500 MPa. During the experiment, the vertical displacements near the failed column P3 were recorded by four LDVT sensors, named P2_11V, P23_1/3V, P23_2/3V and P3_11V. Details of geometry and positions of the LDVTs are shown in Figure 6.

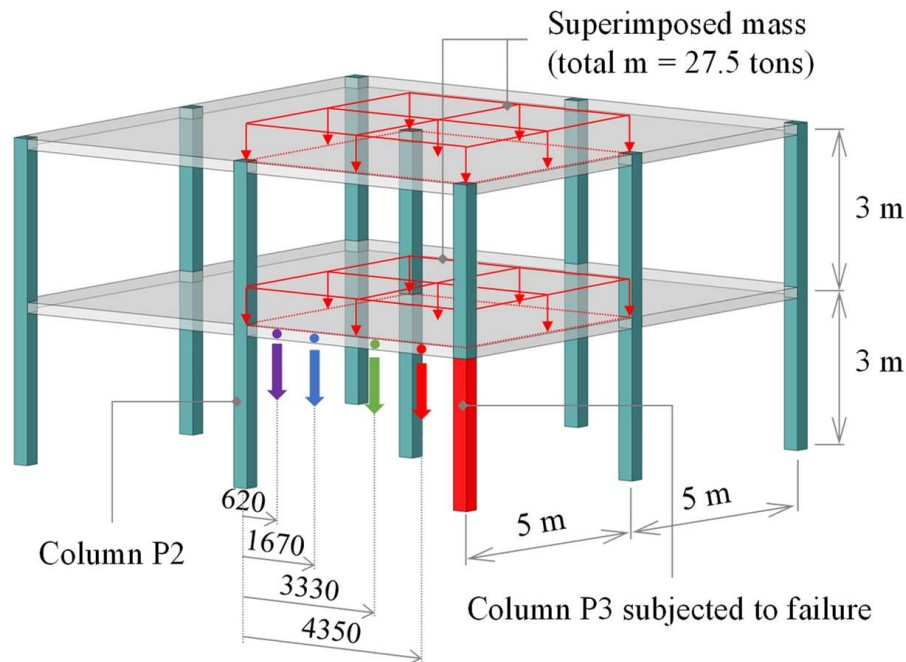

**Figure 6.** Geometry of the two-story reinforced concrete frame with sensors P2_11V (violet), P23_1/3V (blue), P23_2/3V (green), P3_11V (red).

Three-dimensional FE models consist of 8-node solid elements for the concrete and 2-node beam elements for the reinforcement parts. The complete FE model of the two-story RC frame consists of 47,053 solids and 77,218 beams, with an average element dimension equal to 100 mm.

An embedded reinforcement approach provides a perfect bond between reinforcement elements and surrounding concrete material. This approach is implemented using the *CONSTRAINED_BEAM_IN_SOLID [43,44] keyword in LS-DYNA. The bottom faces of the frame columns and slab supports are fixed from any displacements. The detailed FE model overview is shown in Figure 7.

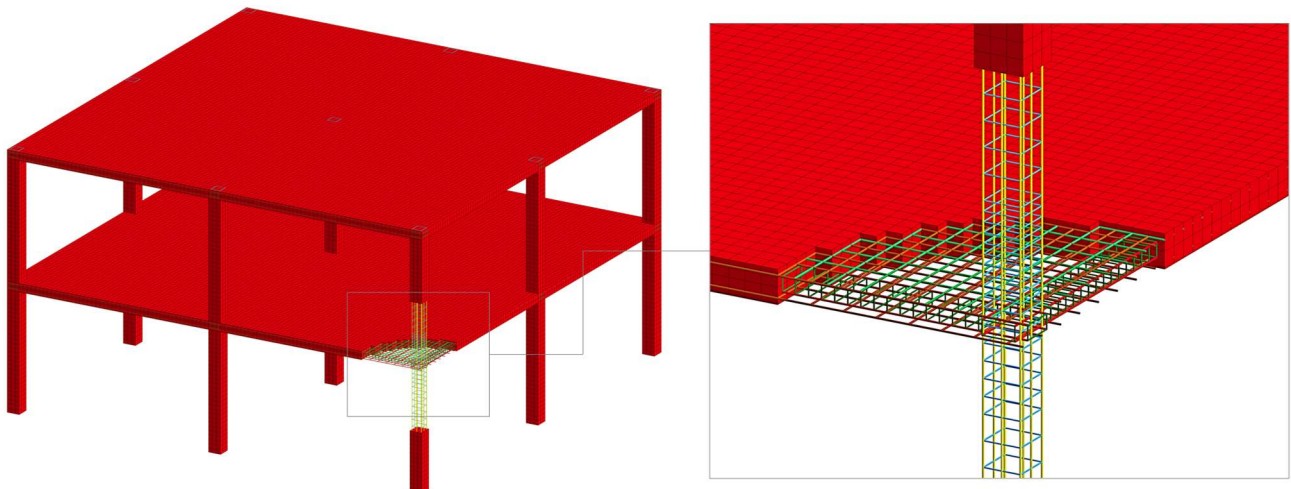

**Figure 7.** Finite element mesh of the two-story frame.

## 3. Results and Discussion

### 3.1. Single Element Strength Estimation

Figure 8 shows the internal model auto calibration results in the concrete strength range of 20–60 MPa. From these data, we can see the peak tensile strength prediction estimation error. The C60 stress–strain curve turns out to be lower than the C50 curve, which is an obvious error (see Figure 8b).

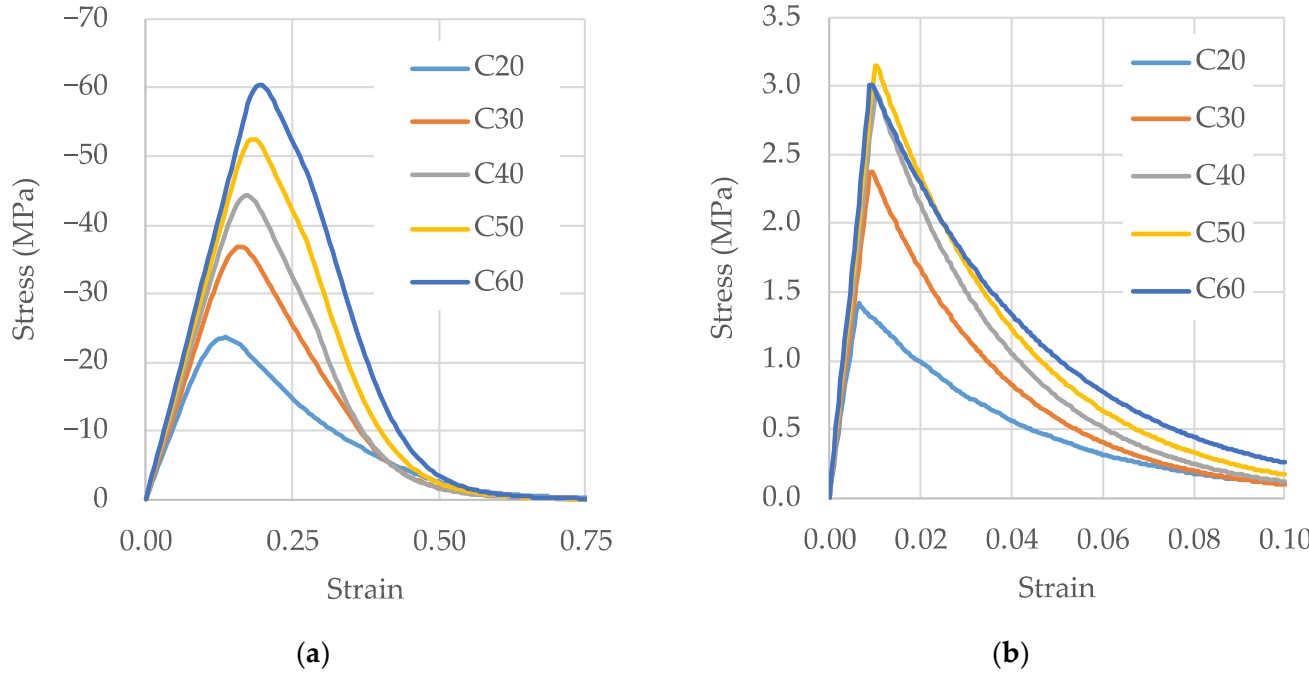

(**a**)                                             (**b**)

**Figure 8.** Single concrete finite element unconfined tests: (**a**) stress-strain relations on compression; (**b**) stress-strain relations on tension.

The single element unconfined tensile and compression testing performed for both auto internal and proposed external calibration procedure and the results of the calculations are shown in Figure 9.

There is significant inaccuracy of up to 56% for tension and 23% for compression in the model performance with auto internal calibration. Table 3 gives the estimate of the inaccuracy. Significant errors in strength estimation persist even in the narrower concrete strength application range of 28–48 MPa, as recommended by the developers [6]. The proposed external calibration significantly increases the accuracy of the material model. The error in strength estimation is reduced to a few percent.

**Table 3.** Strength estimation precision for different CSCM calibration.

| $f_c$ (MPa) | $f_c$ Overestimation (%) | | $f_t$ Underestimation (%) | |
| --- | --- | --- | --- | --- |
| | Auto Internal Calibration | Proposed External Calibration | Auto Internal Calibration | Proposed External Calibration |
| 20–60 | +0.7–+23.0 | +1.5–+5.4 | +18.6–+56.2 | −2.8–+5.8 |
| 30–50 | +5.1–+23.0 | −0.5–+0.6 | +18.6–+56.2 | +0.9–+1.9 |

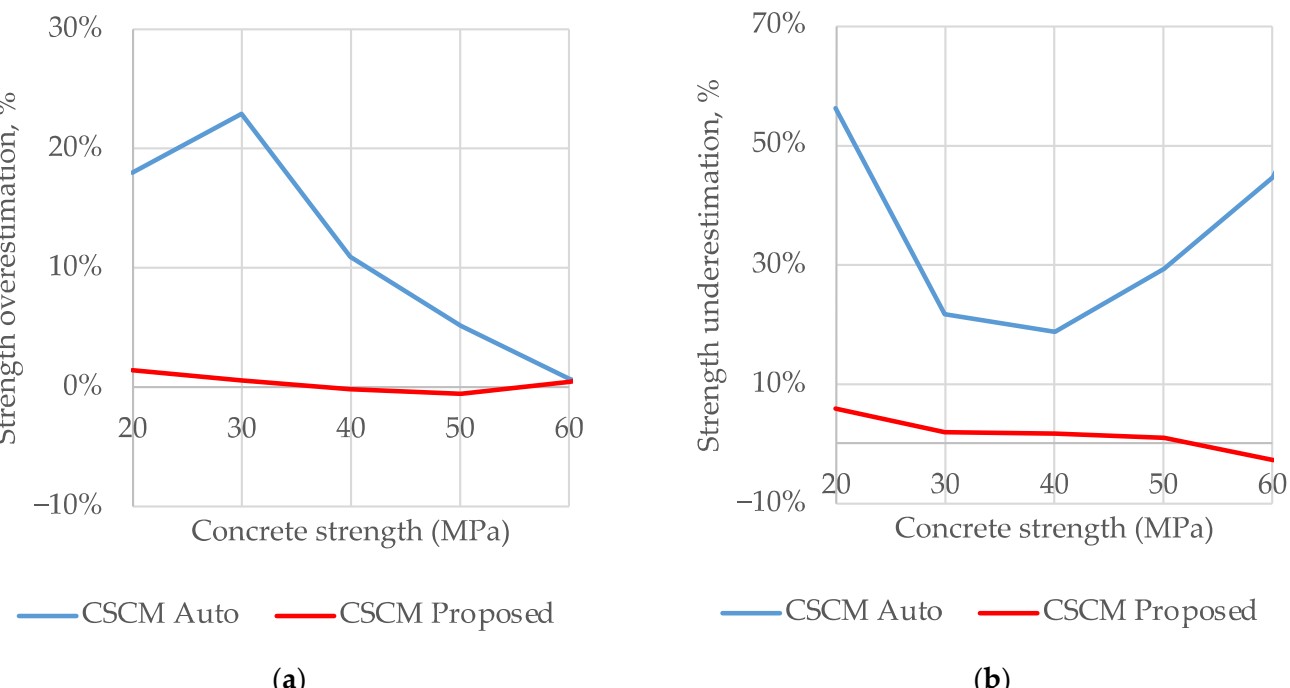

**Figure 9.** CSCM Auto internal and CSCM Proposed external calibration comparison: (**a**) compressive strength; (**b**) tensile strength.

### 3.2. Single Element Fracture Energy Estimation

Figure 10 shows the result of the comparison—fracture energy underestimation—for both auto internal and proposed external calibration procedures.

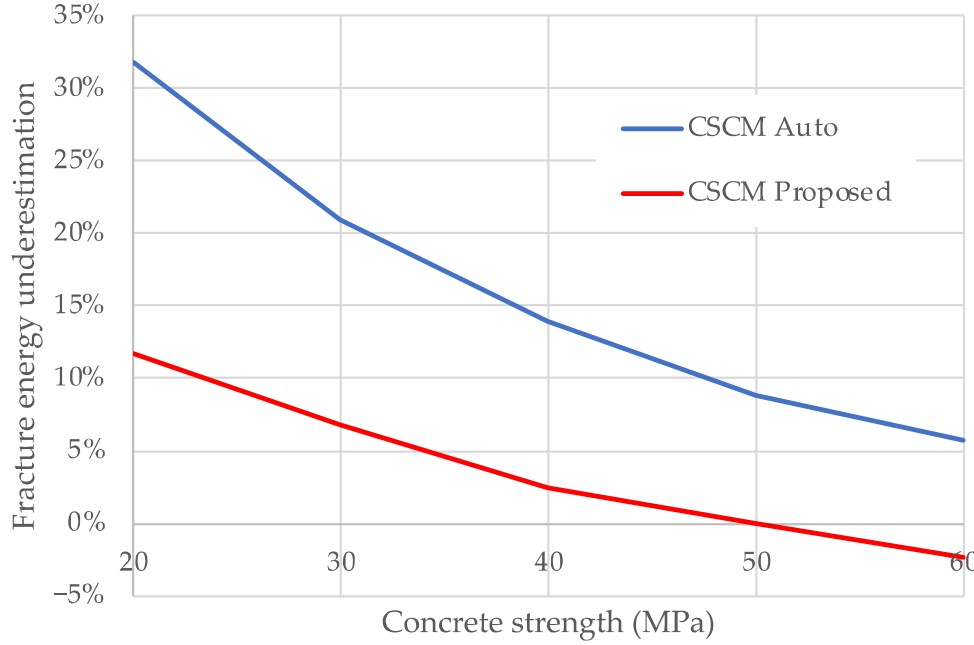

**Figure 10.** CSCM Auto internal and CSCM Proposed external calibration fracture energy comparison.

Table 4 presents in-fracture energy underestimation error summary. A dramatic decrease in the error of the fracture energy border of the extended CSCM concrete model application range with auto internal calibration is observed. It could be seen that new

proposed calibration significantly improves the prediction of fracture energy from [+8.8%; +20.9%] to [+6.8%; 0.0%].

**Table 4.** Fracture energy estimation precision for different CSCM calibration.

| $f_c$ (MPa) | $G_F$ Underestimation (%) | |
|---|---|---|
| | Auto | Proposed |
| 20–60 | +5.8–+31.8 | −2.3–+11.7 |
| 30–50 | +8.8–+20.9 | +6.8–0.0 |

*3.3. Impact on RC Beam*

Figures 11–13 show the unaverage, nonsmoothed visualization of damage field distribution results for the considered three cases simulated with auto internal and proposed external CSCM model calibration procedures.

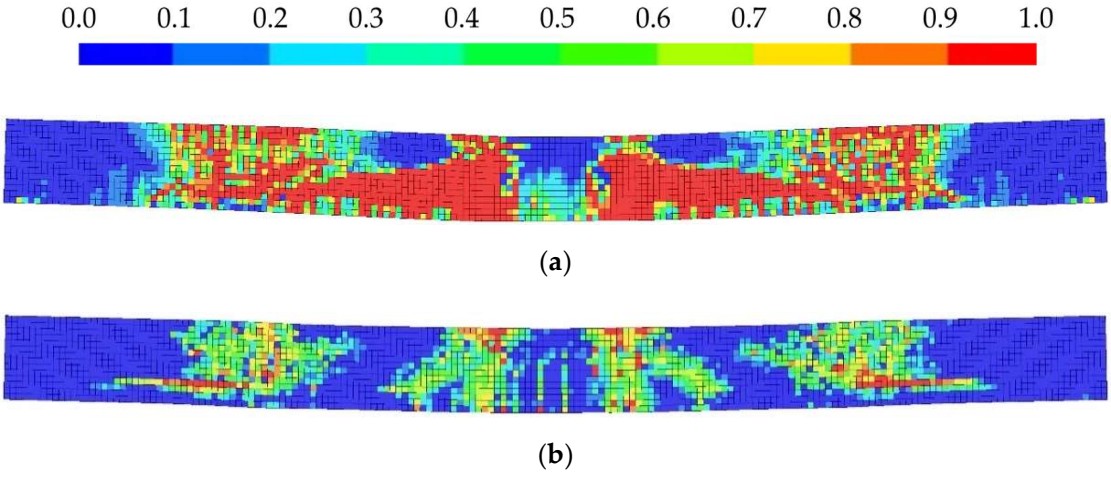

**Figure 11.** Concrete damage in beam after impact Case B: (**a**) CSCM Auto; (**b**) CSCM Proposed.

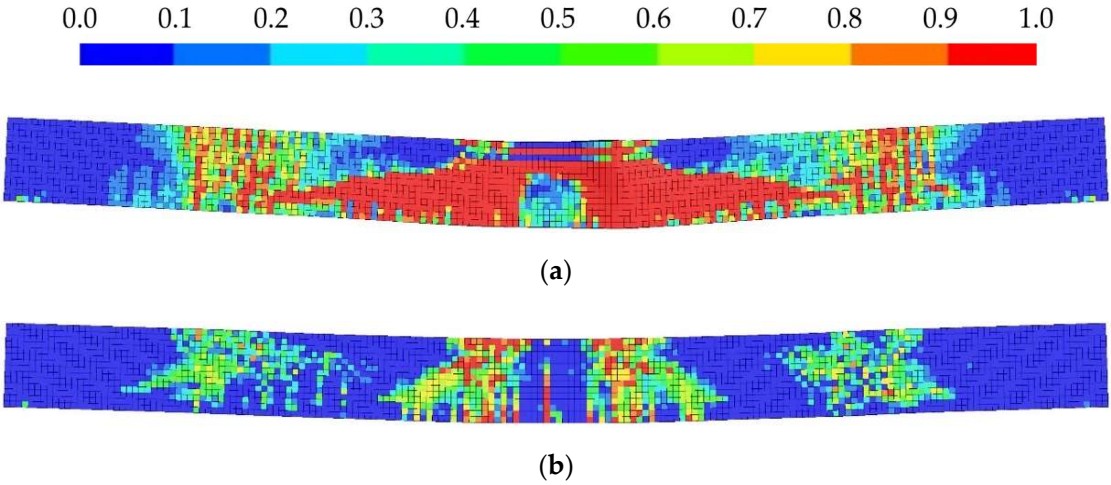

**Figure 12.** Concrete damage in beam after impact Case C: (**a**) CSCM Auto; (**b**) CSCM Proposed.

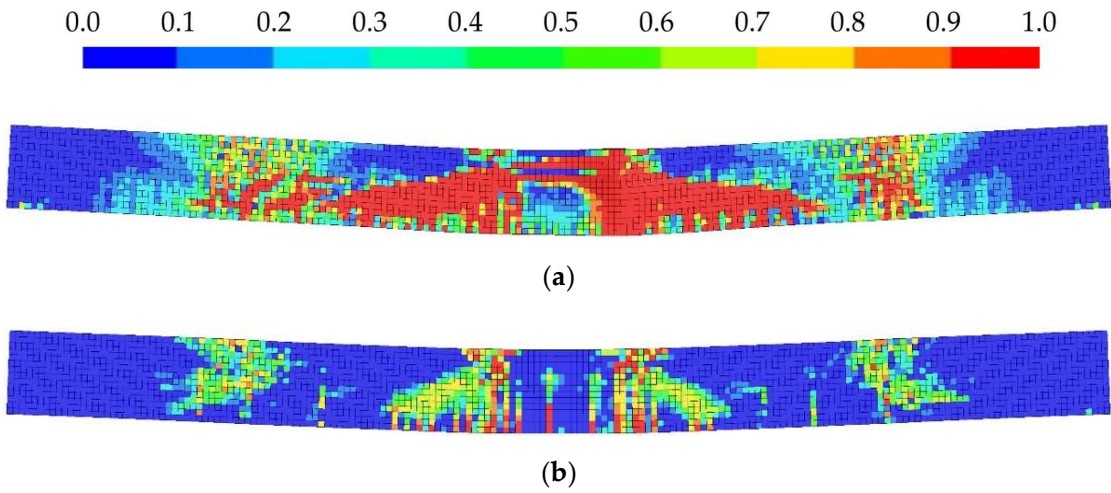

**Figure 13.** Concrete damage in beam after impact Case D: (**a**) CSCM Auto; (**b**) CSCM Proposed.

It could be seen that the total volume of material with maximal damage value is always more significant for the model with internally fitted parameters. The formation of such large areas of damage indicates the presence of zones of material fragmentation rather than single cracks caused by tensile loads.

The results obtained with the proposed calibration procedure show minor damage. They instead look like single cracks in the material, and no crushing is observed.

Detailed images of natural crack patterns in concrete beams for cases B–D are not presented in the Evaluation of LS-DYNA Concrete Material Model 159 manual [23]. Upon impact, multiple cracks primarily initiate on the tensile face of the beam and propagate towards the compressive face.

Figure 14 shows a comparison of experimental and numerical beam deflection history. Peak deflection value and divergence from experiment for different CSCM calibrations is shown in Table 5. The model with the proposed CSCM calibration procedure shows better agreement with the experimental data. Thus, in the most favorable for auto calibration scenario, the new proposed calibration procedure not only does not worsen the results of calculations, but in many respects also improves the qualitative and quantitative description of concrete structures behavior.

**Table 5.** Peak deflection value and divergence from experiments for different CSCM calibration.

| Case | Peak deflection value (mm) | | | Divergence from experiment (%) | |
|---|---|---|---|---|---|
| | Experiment | Auto | Proposed | Auto | Proposed |
| B (31.75 kg, 7.3 m/s) | −24.94 | −28.43 | −22.94 | −14% | 8% |
| C (47.86 kg, 6.0 m/s) | −26.47 | −35.04 | −26.36 | −32% | 0% |
| D (63.93 kg, 5.2 m/s) | −31.25 | −35.60 | −28.50 | −14% | 9% |

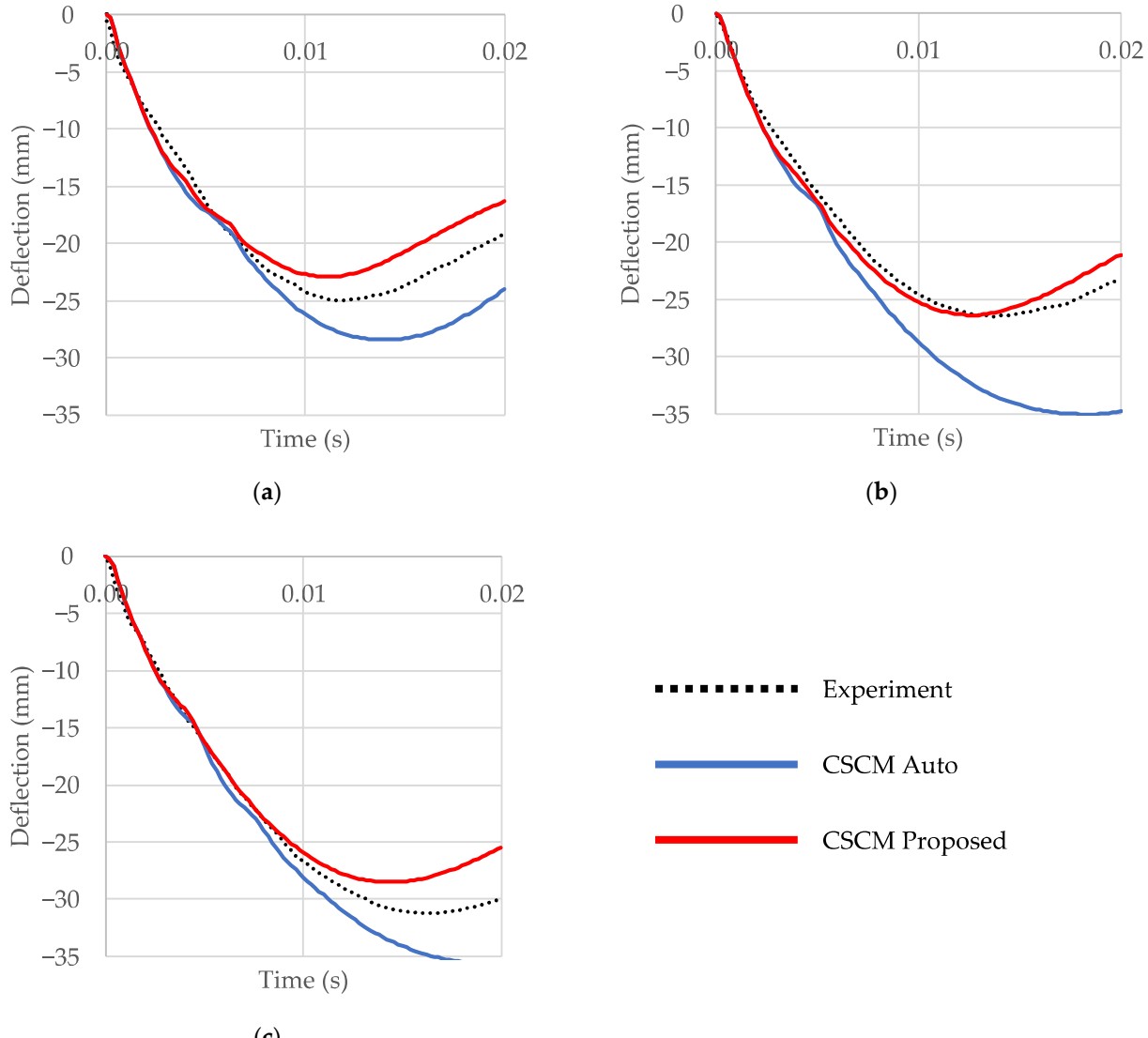

**Figure 14.** CSCM Auto internal and Proposed external calibration comparison: (**a**) case B (31.75 kg, 7.3 m/s); (**b**) case C (47.86 kg, velocity 6.0 m/s); (**c**) case D (63.93 kg, 5.2 m/s).

*3.4. Two-Story Frame Progressive Collapse*

First, the concrete damage field could be studied from the visualization presented in Figure 15. The damage pattern is the expected pattern for both concrete model settings. It can be noted that CSCM Auto shows larger damage zones compared to CSCM Proposed.

Next, let us proceed to the comparison of the model key points displacement history shown in Figure 6. Figure 16 shows the results of the simulation in comparison with the experiment. It could be seen that CSCM Proposed shows excellent agreement with the experiment for all considered points. At the same time, CSCM Auto shows an overestimation of the displacements by almost two times. The peculiarities of the loading conditions explain such unsatisfactory results for CSCM Auto. The two-store frame model uses C30 concrete, which lies on the bottom border of the CSCM Auto accuracy application range. The material expects primarily tensional loading during the frame collapse; this is the regime with the most inaccurate strength and fracture energy estimation for CSCM Auto.

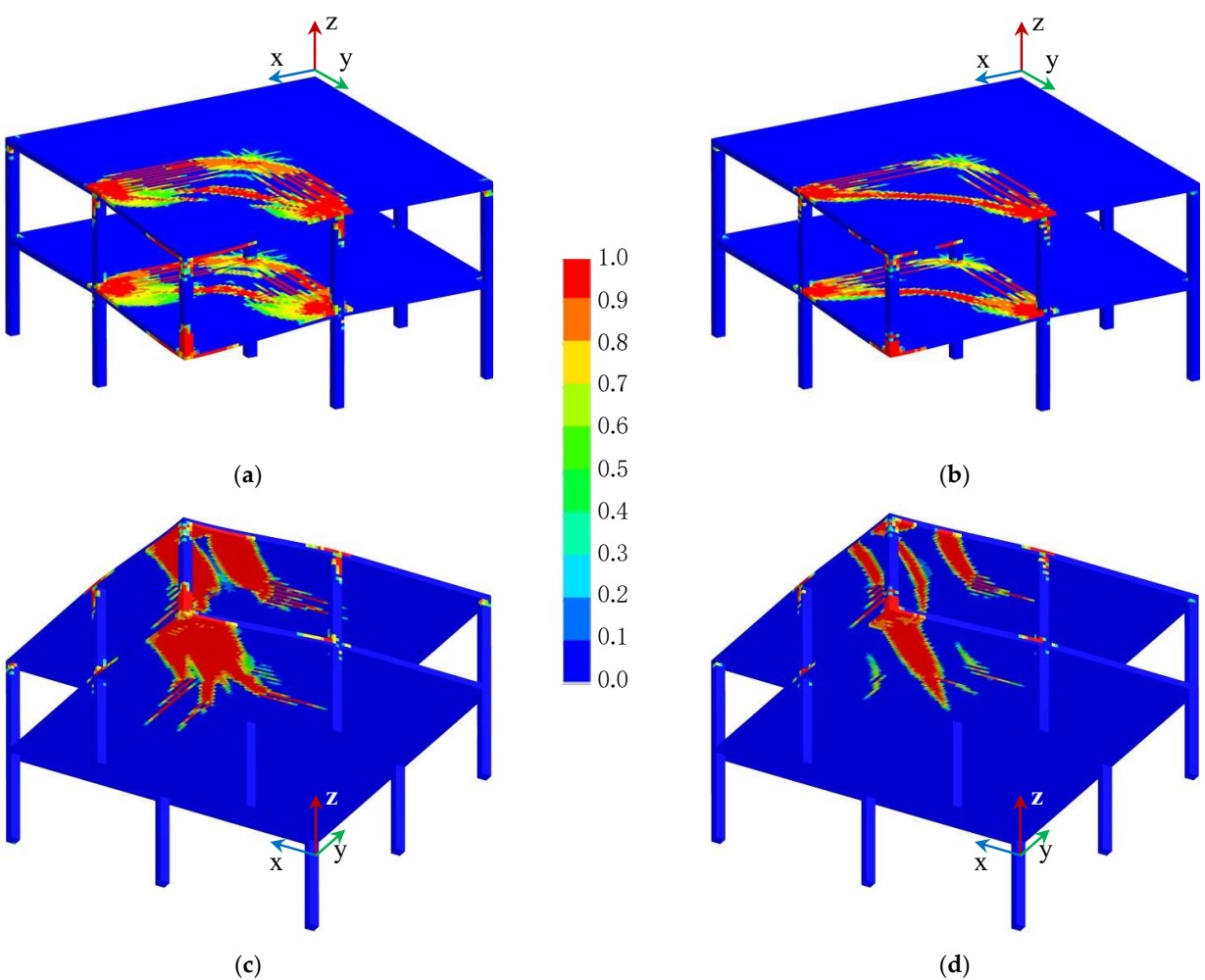

**Figure 15.** Concrete damage pattern: (**a**) CSCM Auto, isometric view; (**b**) CSCM Proposed, isometric view; (**c**) CSCM Auto, isometric bottom view; (**d**) CSCM Proposed, isometric bottom view.

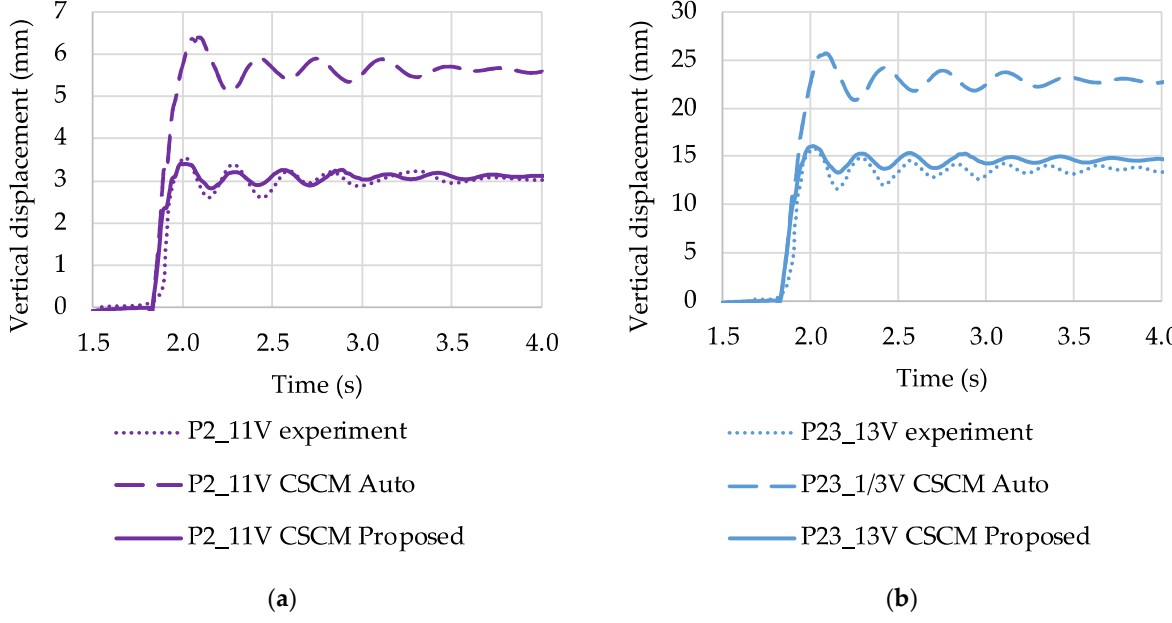

**Figure 16.** *Cont*.

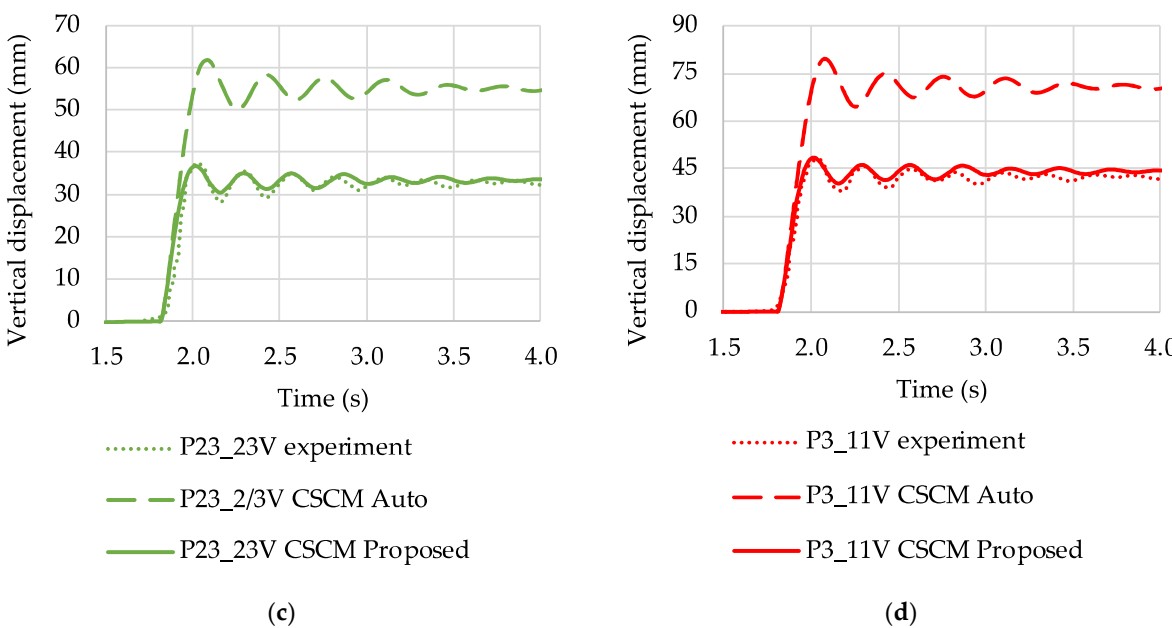

**Figure 16.** Vertical displacements for different points: (**a**) P2_11V; (**b**) P23_1/3V; (**c**) P23_2/3V; (**d**) P3_11V.

## 4. Conclusions

The CSCM model has massive potential for concrete structures simulation under dynamic and static load. Since the automatic adjustment of model parameters in LS-DYNA leads to significant errors, this paper attempts to develop a methodology for calibrating model parameters for concretes of classes C20–C60, most commonly found in civil engineering. The proposed external calibration procedure can significantly improve the qualitative and quantitative description of concrete structure behavior.

Single elements strength studies on default Automatic CSCM model calibration show an overestimation of compressive strength of up to 23.0% and an underestimation of tensile strength up to 56.2%. The fracture energy underestimation is up to 31.8%. The developed calibration procedure reduces these deviations 3.5–10 times: 5.4% on compressive strength, 5.8% on tensile strength and 11.7% on fracture energy.

Due to the lower tensile strength of concrete and low fracture energy, the default Auto CSCM model calibration dramatically underestimates the lifetime of building structures. Two examples of dynamic deformation of RC structures with low loading rates typical for civil structures were considered to validate the calibration procedure.

The first example, the low-velocity impact of a rigid impactor on an RC beam, shows a significant improvement in the detailed description of the fracture process. The crack pattern is more realistic; the peak displacement error for case B decreased from 14% to 8%, for case C, from 32% to 0% and for case D, from 14% to 9%.

The second example is the progressive collapse of a two-story building frame local failure of a corner column. Simulation using the default Auto CSCM parameters calibration, due to significant underestimation of the tensile strength and fracture energy, leads to significant deviations from the experimental data: up to 100% error. Calculations using the proposed parameters of the CSCM model are much better in agreement with the experiment; the errors do not exceed 7%.

Thus, the developed calibration procedure improves the performance of the CSCM model over a range of concrete classes from C20 to C60 at low loading rates. Validation of the proposed procedure at higher loading rates will be the next stage of this study.

**Author Contributions:** Data curation, Y.V.N.; Investigation, Y.V.N. and A.N.D.; Methodology, Y.V.N.; Project administration, D.S.M.; Resources, D.S.M.; Software, Y.V.N.; Supervision, D.S.M.; Visualization, Y.V.N. and A.N.D.; Writing—original draft, Y.V.N.; Writing—review and editing, Y.V.N. All authors have read and agreed to the published version of the manuscript.

**Funding:** The research is partially funded by the Ministry of Science and Higher Education of the Russian Federation under the strategic academic leadership program, 'Priority 2030' (Agreement 075-15-2021-1333 dated 30.09.2021).

**Institutional Review Board Statement:** Not applicable.

**Informed Consent Statement:** Not applicable.

**Acknowledgments:** The authors are grateful to Nikolai Vatin (Peter the Great St. Petersburg Polytechnic University, Russia) for advice and valuable comments while working on this article.

**Conflicts of Interest:** The authors declare no conflict of interest. The funders had no role in the design.

## Appendix A

### Appendix A.1. Concrete Petameters from CEBFIP 1990 Analysis

Some additional material parameters could be estimated with the help of CEBFIP 1990 [9]. The tangent modulus of elasticity assumed as default elasticity modulus since not only elastic concrete analysis is performed:

$$E = E_{c0} \sqrt[3]{\frac{f_c + \Delta f}{f_{cm0}}}, \tag{A1}$$

where: $E_{c0} = 21.5 \cdot 10^3$ MPa, $\cdot f = 8$ MPa, $f_{cm0} = 10$ MPa. Poison ratio assumed to be constant for all rage of applicability $\nu = 0.2$.

Modulus of elasticity [9].

Shear modulus and bulk modulus could be estimated as:

$$G = \frac{E}{2(1+\nu)} \tag{A2}$$

$$K = \frac{E}{3(1-2\nu)} \tag{A3}$$

Optional kinematic hardening could enable hardening initiation and hardening rate parameters set no equivale to zero [23]. Prepeak nonlinearity is more pronounced in compression than in tension or shear. This equation is optional and not default because it is not essential to the good performance of the model.

$$NH = 0 \tag{A4}$$

$$CH = 0 \tag{A5}$$

$NH$ parameters could be estimated in case of kinematic hardening for compression, to be taken into account [45] as:

$$NH = \frac{f_c^{0.855}}{60}. \tag{A6}$$

A second parameter, $CH$, determines the rate of hardening (amount of nonlinearity) and could be fitted with a single element compression test.

### Appendix A.2. Parameters for Compressive Meridian (TXC)

The compressive meridian curve for the strength surface of the concrete is given in the form:

$$\text{TXC} = F_f(I_1) = \alpha + \theta I_1 - \lambda e^{-\beta I_1}, \tag{A7}$$

where $\alpha$, $\theta$, $\lambda$, $\beta$ are direct input parameters of *MAT_CSCM, according to Table A1.

**Table A1.** TXC parameters.

| *MAT_CSCM Variable | Units | Equation |
|---|---|---|
| ALPHA | MPa | $\alpha = 13.9846 e^{\left(\frac{f_c}{68.8756}\right)} - 13.8981$ |
| THETA | - | $\theta = 0.3533 - 3.3294 \cdot 10^{-4} f_c - 3.8182 \cdot 10^{-6} f_c^2$ |
| LAMBDA | MPa | $\lambda = 3.6657 e^{\frac{f_c}{39.9363}} - 4.7092$ |
| BETA | MPa$^{-1}$ | $\beta = 18.17791 f_c^{-1.7163}$ |

*Appendix A.3. Parameters for Shear Meridian (TOR)*

The shear meridian curve for the strength surface of the concrete is given in the form [35]:

$$TOR = Q_1(I_1) F_f(I_1), \ Q_1(I_1) = \alpha_1 + \theta_1 I_1 - \lambda_1 e^{-\beta_1 I_1}, \tag{A8}$$

where $\alpha_1$, $\theta_1$, $\lambda_1$, $\beta_1$ is direct input parameters of *MAT_CSCM, according to Table A2.

**Table A2.** TOR parameters.

| *MAT_CSCM Variable | Units | Equation |
|---|---|---|
| ALPHA1 | - | $\alpha_1 = 0.82$ |
| THETA1 | MPa$^{-1}$ | $\theta_1 = 0$ |
| LAMBDA1 | - | $\lambda_1 = 0.2407$ |
| BETA1 | MPa$^{-1}$ | $\beta_1 = 0.33565 f_c^{-0.95383}$ |

*Appendix A.4. Parameters for Tensile Meridian (TXE)*

The shear meridian curve for the strength surface of the concrete is given in the form [35]:

$$TOR = Q_2(I_1) F_f(I_1), \ Q_2(I_1) = \alpha_2 + \theta_2 I_1 - \lambda_2 e^{-\beta_2 I_1}, \tag{A9}$$

where $\alpha_2$, $\theta_2$, $\lambda_2$, $\beta_2$ is direct input parameters of *MAT_CSCM, according to Table A3.

**Table A3.** TXE parameters.

| *MAT_CSCM Variable | Units | Equation |
|---|---|---|
| ALPHA2 | - | $\alpha_2 = 0.76$ |
| THETA2 | MPa$^{-1}$ | $\theta_2 = 0$ |
| LAMBDA2 | - | $\lambda_2 = 0.26$ |
| BETA2 | MPa$^{-1}$ | $\beta_2 = 0.285 f_c^{-0.94843}$ |

*Appendix A.5. Cap Surface Parameters*

Direct input parameters of *MAT_CSCM for cap surface could be found in Table A4 [35].

**Table A4.** Cap surface parameters.

| *MAT_CSCM Variable | Units | Equation |
|---|---|---|
| R | - | $R = 4.45994e^{-\frac{f_c}{11.51679}} + 1.95358$ |
| X0 | MPa | $X_0 = 17.087 + 1.892 f_c$ |
| W | - | $W = 0.065$ |
| D1 | MPa | $D_1 = 6.110 \cdot 10^{-4}$ |
| D2 | MPa$^2$ | $D_2 = 2.225 \cdot 10^{-6}$ |

*Appendix A.6. Damage and Energy Parameters*

Direct input parameters of *MAT_CSCM for damage and fracture could be found in Table A5 [6].

**Table A5.** Damage and energy parameters.

| *MAT_CSCM Variable | Units | Equation |
|---|---|---|
| B | - | $B = 100$ |
| D | - | $D = 0.1$ |
| GFC | MPa·mm | $G_{FC} = 100 \cdot G_F$ |
| GFT | MPa·mm | $G_{FT} = G_F$ |
| GFS | MPa·mm | $G_{FS} = G_F$ |
| PWRC | - | $pwrc = 5$ |
| PWRT | - | $pwrt = 1$ |
| PMOD | - | $pmod = 0$ |

*Appendix A.7. Strain Rate Parameters*

Direct input parameters of *MAT_CSCM for strain rate effects consideration could be found in Table A6 [6].

**Table A6.** Strain Rate Parameters.

| *MAT_CSCM Variable | Units | Equation |
|---|---|---|
| ETA0C | - | $\eta_{0c} = 1.2772337 \cdot 10^{-11} \cdot f_{c_{psi}}^2 - 1.0613722 \cdot 10^{-7} \cdot f_{c_{psi}} + 3.203497 \cdot 10^{-4}$ |
| NC | - | $\eta_c = 0.78$ |
| ETA0T | - | $\eta_{0t} = 8.0614774 \cdot 10^{-13} \cdot f_c^2 + -9.77736719 \cdot 10^{-10} \cdot f_c + 5.0752351 \cdot 10^{-5}$ |
| NT | - | $\eta_t = 0.48$ |
| OVERC | - | $owerc = 1.309663 \cdot 10^{-2} \cdot f_c^2 - 0.3927659 \cdot f_c + 21.45$ |
| OVERT | - | $overt = 1.309663 \cdot 10^{-2} \cdot f_c^2 - 0.3927659 \cdot f_c + 21.45$ |
| SRATE | - | $Srate = 1$ |
| REPOW | - | $repow = 1$ |

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
