# Peer review of "Precise Calibration of the Continuous Surface Cap Model for Concrete Simulation"

_buildings, doi:10.3390/buildings12050636_

Round 1

Reviewer 1 Report

This paper focused on the the numerical simulation of concrete using the CSCM model. The workload of the study is enough and the the subject of the study meets the requirements of the journal. However, before publication, the manuscript needs to be revised according to the following opinions.

  1. Section 1, it is recommended to discuss the differences between the CSCM model and the similar material models, like HJC (#111) model and KCC (#72R3) model. The discussions can make the readers more clearly understand the advantages of CSCM model.
  2. It is recommended to add some new papers about other material models in the introduction paper, like

            https://doi.org/10.1016/j.jobe.2020.101610

  1. Section 2.3, some diagrams are needed to illustrate the process of the single-element model.
  2. Section 3, it is necessary to discuss convergence of mesh size.
  3. Figure 7, some reference curves from other papers need to be added to verify the numerical results.
  4. It is suggested to add a partially enlarged view of the numerical results in Figure 13 for comparison with the experimental results.

Author Response

Dear reviewer, thank you for your time and valuable comments on our manuscript. They have helped identify its shortcomings and significantly improve the quality of the material.

Reviewer 2 Report

The present paper calibrated the continuous surface cap model in LS-DYNA for concrete simulation. The topic is interesting. However, the writing of the paper needs improvement. The article can only be accepted for publication in the journal of Buildings after the following questions are answered or addressed appropriately:

  1. Page 4, Line 149, “The meridional surface in the Principal Stress Space (PSS) is shown in the Figure 1a.” Figure 1a is not the meridional surface. Page 5, Line 174 “The elliptical surface in the PSS is shown in Figure 1b.” is not correct. Figure 1b does not show any elliptical surface. Figure 1 needs to be revised accordingly.
  2. Sections 2.1 and 2.2 are not original works. The introduction of the previously available model is lengthy.
  3. Page 7, Line 246, “cup” should be “cap”.
  4. Page 9, Line 319, why 8.5 mm hex elements are adopted? The mesh size sensitivity analysis needs to be done.
  5. L427, Figures 15(c) and (d) need to be rotated slightly to make the figures clearer.

Author Response

(The authors gave the same response as above.)

Round 2

Reviewer 2 Report

It is recommended to modify the size of Figure 3 to make it look more appropriate.